# Transient Receptor Potential Vanilloid 1 Signaling Is Independent on Protein Kinase A Phosphorylation of Ankyrin-Rich Membrane Spanning Protein

**DOI:** 10.3390/medsci10040063

**Published:** 2022-11-17

**Authors:** Antonio Pellegrino, Sandra Mükusch, Viola Seitz, Christoph Stein, Friedrich W. Herberg, Harald Seitz

**Affiliations:** 1Fraunhofer Institute for Cell Therapy and Immunology, 14476 Potsdam, Germany; 2Institute of Experimental Anaesthesiology, Charité—Universitätsmedizin Berlin, 12203 Berlin, Germany; 3Brandenburg Medical School Theodor Fontane, Fehrbelliner Str. 38, 16816 Neuruppin, Germany; 4Department of Biochemistry, University of Kassel, 34132 Kassel, Germany

**Keywords:** TRPV1, ARMS, PKA, AKAP79, pain, phosphorylation

## Abstract

The sensory ion channel transient receptor potential vanilloid 1 (TRPV1) is mainly expressed in small to medium sized dorsal root ganglion neurons, which are involved in the transfer of acute noxious thermal and chemical stimuli. The Ankyrin-rich membrane spanning protein (ARMS) interaction with TRPV1 is modulated by protein kinase A (PKA) mediating sensitization. Here, we hypothesize that PKA phosphorylation sites of ARMS are crucial for the modulation of TRPV1 function, and that the phosphorylation of ARMS is facilitated by the A-kinase anchoring protein 79 (AKAP79). We used transfected HEK293 cells, immunoprecipitation, calcium flux, and patch clamp experiments to investigate potential PKA phosphorylation sites in ARMS and in ARMS-related peptides. Additionally, experiments were done to discriminate between PKA and protein kinase D (PKD) phosphorylation. We found different interaction ratios for TRPV1 and ARMS mutants lacking PKA phosphorylation sites. The degree of TRPV1 sensitization by ARMS mutants is independent on PKA phosphorylation. AKAP79 was also involved in the TRPV1/ARMS/PKA signaling complex. These data show that ARMS is a PKA substrate via AKAP79 in the TRPV1 signaling complex and that all four proteins interact physically, regulating TRPV1 sensitization in transfected HEK293 cells. To assess the physiological and/or therapeutic significance of these findings, similar investigations need to be performed in native neurons and/or in vivo.

## 1. Introduction

The transient receptor potential vanilloid 1 (TRPV1) was initially discovered as heat- and capsaicin-activated ion channel involved in mediating acute nociceptive signaling [1,2]. It is mainly expressed in primary sensory (nociceptive) neurons and is involved in thermosensation in the pathogenesis of inflammatory and neuropathic pain and persistent itch [3,4,5]. TRPV1 is activated by a wide array of noxious chemical and physical stimuli like capsaicin, resiniferatoxin, protons, endovanilloids, and temperatures above 43 °C. TRPV1 is modulated by phospholipases, phospholipids, kinases, phosphatases, and cAMP targets [6,7,8]. Further research shifted the focus to its involvement in chronic pain [9].

Several sensitizing membrane-associated signaling complexes and modulating interaction partners of TRPV1 are known in nociceptive neurons. The A-kinase anchoring protein 79 (AKAP79 in humans, AKAP150 in rodents) was found to be an important sensitization-mediating interaction partner of TRPV1. It facilitates the binding of protein kinase A (cAMP-dependent protein kinase, PKA) to its substrates by tethering it to specific subcellular sites, resulting in phosphorylation-induced TRPV1 sensitization [10,11,12,13].

Recently, it was shown that the Ankyrin-rich membrane spanning protein (ARMS), also known as Kinase D Interacting Substrate 220 (KIDINS220), is another potential PKA-dependent sensitization-mediating interaction partner of TRPV1 in mouse dorsal root ganglion (DRG) neurons [14]. The actual mechanism of ARMS-mediated PKA-dependent TRPV1 sensitization is not known. Therefore, we aimed to investigate potential PKA phosphorylation sites of ARMS. PKA targets a (**R**/K)-(**R**/K)-X-(**S**/T) ψ recognition sequence, where the bold letters code for preferential sites and ψ for any hydrophobic amino acid [15]. ARMS was identified as a 220 kDa large adaptor protein harboring four putative transmembrane domains, multiple ankyrin repeats, a sterile alpha motif domain and a potential PDZ-binding motif [16,17]. It is specifically expressed in the developing nervous system as well as in highly plastic areas of the adult brain and appears to be involved in neuronal differentiation, survival, and synaptic plasticity [18]. Additionally, ARMS was identified as the first physiological protein kinase D (PKD) substrate [16].

Since the PKA recognition sequence shares similarities with the PKD recognition sequence (L/V/I)-X-(R/K)-X-X-(S/T), experiments were done to discriminate between PKA and PKD [19]. Here, we explore the functional and sensitizing role of ARMS mutants without PKA phosphorylation sites in the context of TRPV1 activation in transfected cells.

## 2. Materials and Methods

### 2.1. Cell Culture

Human embryonic kidney (HEK) 293 cells (DSMZ, Braunschweig, Germany) were maintained at 37 °C, 5% CO_2_ in DMEM (Biochrom, Berlin, Germany) containing 10% fetal bovine serum (Biochrom, Berlin, Germany) without antibiotics. They were passaged 1:6–1:12 every second to third day, not exceeding 80% confluence.

### 2.2. Transfection

HEK293 cells were plated at a density of 8 × 10^5^/6 well and 24 h later transiently transfected with the following vectors: pcDNA3.1-TRPV1-yellow fluorescent protein (YFP) (0.5 μg/6 well) and/or pcDNA3.1-ARMS (2.5 µg/6 well) and/or plasmid red fluorescence protein-chloramphenicol-puromycin resistance-short hairpin (pRFP-C-RS-sh; 1.75 µg/6 well) using X-tremeGENE HP DNA Transfection Reagent (Roche, Mannheim, Germany) in a 3:1 ratio. All plasmids are listed in 2.9. Plasmids and Peptides. For electrophysiological experiments, cells were cultured and transiently transfected as described previously [14]. For PKA activation, cells were washed after 48 h, incubated in DMEM without fetal calf serum (FCS) for 2 h and stimulated with 50 µM forskolin together with 100 µM IBMX. For PKA/PKD inhibition, cells were washed after 48 h, incubated in DMEM without FCS for 2 h and 10 µM H89 or 1 µM BPKDi. Subsequently, cells were washed once with cold PBS containing 1 mM sodium orthovandate (Sigma-Aldrich Chemie GmbH, Taufkirchen, Germany) and lysed using RIPA Lysis and Extraction Buffer with Halt Protease and Phosphatase Inhibitor Cocktail (Thermo Scientific, Darmstadt, Germany). Cell debris was removed by centrifugation at 13,000× *g* for 10 min at 4 °C and protein concentrations were measured using Pierce BCA Protein Assay Kit (Thermo Scientific, Darmstadt, Germany).

### 2.3. Immunoprecipitation

Equal protein concentrations of the cleared cell lysates were used for immunoprecipitation. Each sample was incubated overnight and end-to-end mixing with 10^4^ anti-ARMS (mouse, MA1-90667, Invitrogen, Darmstadt, Germany) coupled MagPlex Microspheres (Luminex Corp., Hertogenbosch, The Netherlands). Incubated MagPlex Microspheres were washed twice with PBST and incubated for 1 h at 900 rpm with anti-ARMS (rabbit, ab34790, abcam, Berlin, Germany), anti-TRPV1 (rabbit, ACC-030, Alomone Labs, Jerusalem, Israel), anti-AKAP79 (rabbit, D28G3, Cell Signaling, Leiden, The Netherlands), or anti-Phospho-PKA substrate (rabbit, 100G7E, Cell Signaling, Leiden, The Netherlands). After antibody incubation, MagPlex Microspheres were washed twice with PBST, incubated for 1 h at 900 rpm with anti-Rabbit-PE (goat, P-2771MP, Invitrogen, Darmstadt, Germany) and washed twice with PBST again. MagPlex Microspheres were resuspended in PBST and analyzed via FLEXMAP 3D (Luminex Corp., Hertogenbosch, The Netherlands).

### 2.4. Western Blot Analysis

For Western Blot analysis, equal protein concentrations of the cleared cell lysates were used for Protein G Mag Sepharose immunoprecipitation. First, 25 μL Protein G Mag Sepharose (Cytiva, Marlborough, MA, USA) were washed twice with RIPA Lysis and Extraction Buffer with Halt Protease and Phosphatase Inhibitor Cocktail (Thermo Scientific, Darmstadt, Germany) and incubated 4 h at 4 °C with 1 µg anti-ARMS (mouse, MA1-90667, Invitrogen, Darmstadt, Germany) by end-to-end mixing. Antibody-incubated Protein G Mag Sepharose was washed twice, incubated overnight with 1 mg cleared cell lysate by end-to-end mixing and washed twice. Samples were eluted with 30 µL Sample Buffer, Laemmli 2× Concentrate (Sigma-Aldrich Chemie GmbH, Taufkirchen, Germany) at 37 °C for 30 min and subjected to SDS-PAGE using 8% polyacrylamide gels. After separation, proteins were transferred to a Immobilon-P PVDF membrane (Merck Millipore, Darmstadt, Germany). The Membrane was blocked with 5% BSA (Carl Roth GmbH + Co. KG, Karlsruhe, Germany) in PBST for 1 h at room temperature. Afterwards, the membrane was probed with the following primary antibodies overnight at 4 °C: 1:1000 anti-ARMS (rabbit, ab34790, abcam, Berlin, Germany), 1:1000 anti-TRPV1 (rabbit, ACC-030, Alomone Labs, Jerusalem, Israel), and 1:1000 anti-AKAP79 (rabbit, D28G3, Cell Signaling, Leiden, The Netherlands). Following three washing steps with PBST, the blot was incubated with 1:1000 anti-Rabbit Alexa Fluor 555 (goat, A-21428, Invitrogen, Darmstadt, Germany) secondary antibody for 1 h at room temperature and washed thrice. Protein bands were visualized using Amersham Typhoon Biomolecular Imager (GE Healthcare, Braunschweig, Germany).

### 2.5. Phospho-PKA Substrate Screening

ARMS peptides containing possible PKA-sites (Peps 4LS GmbH, Heidelberg, Germany) were coupled with MagPlex Microspheres (Luminex Corp., Hertogenbosch, The Netherlands). All peptides are listed in 2.9. Plasmids and Peptides. Coupled MagPlex Microspheres were incubated 30 min at 30 °C and 900 rpm with 125 nM PKA catalytic subunit (Biaffin GmbH & Co KG, Kassel, Germany), 0.6 mM ATP in 50 mM TRIS HCl (pH 7.7), and 10 mM MgCl2. MagPlex Microspheres were washed twice with PBST and incubated for 1 h at 900 rpm with anti-Phospho-PKA substrate antibody (rabbit, 100G7E, Cell Signaling, Leiden, The Netherlands). After antibody incubation, MagPlex Microspheres were washed twice with PBST, incubated for 1 h at 900 rpm with anti-Rabbit-PE (goat, P-2771MP, Invitrogen, Darmstadt, Germany), and washed twice with PBST again. Subsequently, MagPlex Microspheres were resuspended in PBST and analyzed via FLEXMAP 3D (Luminex Corp., Hertogenbosch, The Netherlands).

### 2.6. Calcium Flux Measurements

HEK293 cell were plated at a density of 4 × 10^4^ cells /96 well and simultaneously transiently transfected with the following vectors: pcDNA3.1-TRPV1-YFP (0.5 μg/30 wells) and/or with pcDNA3.1-ARMS (2.5 µg/30 wells) and/or pRFP-C-RS-sh (1.75 µg/30 wells) using X-treme GENE HP DNA Transfection Reagent (Roche, Mannheim, Germany) in a 3:1 ratio. At 48 h post transfection, HEK 293 cells were washed and loaded with the fluorescent calcium indicator dye Fluo 4 AM (1 μM, Invitrogen, Darmstadt, Germany) in DMEM without phenol red for 60 min. Baseline calcium flux was measured with FLUOstar^®^ Omega Microplate Reader (BMG LABTECH, Ortenberg, Germany): bottom reading, 7 × 7 well scanning, 485 BP/510, orbital shaking and at 37 °C. TRPV1 was activated by capsaicin (0.1 nM–10000 nM) and calcium flux was measured again with FLUOstar^®^ Omega Microplate Reader (BMG LABTECH, Ortenberg, Germany).

### 2.7. Electrophysiology

Whole-cell voltage-clamp recordings were performed in TRPV1 or TRPV1 and wildtype or mutant ARMS-expressing HEK 293 cells. Cells were held at −60 mV holding potential using an EPC-10 patch clamp amplifier and the PULSE software (HEKA Elektronik, Lambrecht, Germany) as described previously [6,7,14,20]. Experiments were performed 48 h after transfection and the TRPV1 channel was activated by the addition of 50 nM capsaicin.

### 2.8. Statistics

Data are presented as means ± SEM. N is defined as number of biological replicates. Statistical analysis was performed by using 2-way ANOVA with Dunnett’s multiple comparisons test. Statistical significance is denoted as *p* < 0.05 (*), *p* < 0.01 (**), *p* < 0.001 (***), *p* < 0.0001 (****). Outliers were identified and removed with ROUT method (Q = 1%). For patch clamp experiments, data were tested for normal distribution using the Shapiro–Wilk test and analyzed using the Kruskal–Wallis test with Dunn’s multiple comparison test. Dose–response curves were fitted with the Hill equation. All tests were performed using Prism 8.4.3 and 9.1.2 (GraphPad, San Diego, CA, USA).

### 2.9. Plasmids and Peptides

Plasmids and peptides used in the experiments are listed in Table 1. and Table 2. Each plasmid encoding for ARMS mutant, only contained its denoted single PKA-site, while all other PKA-sites were mutated replacing serine or threonine with an alanine. In each peptide, the starting H is used to couple the peptides site-directed to the beads. The GKPIPNPLLGLDST (V5-tag) in every peptide is used to control the said coupling, without the need of a peptide specific antibody. The remaining amino acids are ARMS specific with the potential PKA-site in the center. Potential PKA sites are marked in red.

## 3. Results

Initially, potential PKA-sites in the sequence of ARMS [22] were identified using a (**R**/K)-(**R**/K)-X-(**S**/T) pattern search. This revealed five canonical PKA-sites: S882 (RRVSQ), T903 (RRDTY), S1251/1252 (RRSSH), S1439/40 (KKSSE) and S1526/27 (KKDSSD). The respective peptides and alanine mutants (serine and threonine were exchanged with alanine) were synthetized.

### 3.1. PKA-Sites in ARMS

To screen for phosphorylation of putative PKA-sites, the above-described peptides were coupled to MagPlex Microspheres (Luminex Corp., Hertogenbosch, The Netherlands), incubated with PKA and analyzed by the Luminex FlexMap3D platform (Luminex Corp., Hertogenbosch, The Netherlands) (Figure 1). PKA sites, including two serines (S1251/52, S1439/40, and S1526/27), were determined via single serine to alanine mutants of the corresponding peptide. Phosphorylation was detected using a Phospho-PKA substrate antibody (rabbit, 100G7E, Cell Signaling, Leiden, The Netherlands). This screening revealed highly phosphorylatable T903 and S1251/52 as well as low phosphorylatable S1526/27. The quantification of the phosphorylation level of the double serine sites showed that S1252 is the favored serine in S1251/52. The double phosphorylation site S1526/27 showed an equal phosphorylation of both sites. In addition, the phosphorylation of PKA-sites harboring a lysine motive (S1439/40 and S1526/27) were more dependent on time than arginine containing motives (S882, T903, and S1251/52). Thus, lysine harboring motives reached a phosphorylation maximum after 10 min, while phosphorylation of arginine-containing motives was continuously increasing over a period of 30 min under the chosen conditions.

### 3.2. ARMS/TRPV1 Co-Immunoprecipitation

To investigate potential functional and physical interactions between the discovered phosphorylation sites of ARMS and TRPV1, TRPV1 and PKA-site-deficient ARMS mutants were co-expressed in HEK293 cells. Transfection efficiency of PKA-site mutants and wild type ARMS remained the same. Each ARMS mutant only contained its denoted single PKA-site, while all other PKA-sites were mutated replacing serine or threonine with an alanine. Co-immunoprecipitation experiments were performed with an antibody directed against ARMS (mouse, MA1-90667, Invitrogen, Darmstadt, Germany). The precipitate was further analyzed by the Luminex FlexMap3D platform (Luminex Corp., Hertogenbosch, The Netherlands) with antibodies against TRPV1 (rabbit, ACC-030, Alomone Labs, Jerusalem, Israel), ARMS (rabbit, ab34790, abcam, Berlin, Germany) and a phospho-PKA substrate antibody (rabbit, 100G7E, Cell Signaling, Leiden, The Netherlands). For determination of the native interaction rate of TRPV1 and ARMS, the transfected HEK293 cells did not undergo any additional treatments. To characterize the influence of PKA phosphorylation in terms of a decreasing or increasing interaction rate, transfected HEK293 cells were additionally treated with the PKA inhibitor H89 (10 µM), the PKD inhibitor BPKDi (1 µM), or forskolin in combination with IMBX (50 µM/100 µM, PKA stimulation). The results were then transformed into percent of TRPV1 and ARMS interaction by dividing the TRPV1 mean fluorescence intensity (MFI) with the ARMS MFI for each mutant as well as the wild type (Figure 2).

The known interaction between TRPV1 and wild type ARMS was confirmed by the screening. In addition, it showed differences for the ARMS mutants. About 9% of the wild type ARMS signal was associated with TRPV1’s signal, while 5% of ARMS_0_, 10% of ARMS_S882_, 12% of ARMS_T903_, 15% of ARMS_S1251/52_, 7% of ARMS_S1439/40_, and 60% of ARMS_S1526/27_ were correlated with a TRPV1 signal. The relative MFI ranged in a narrow window between 5–15% for all mutants except ARMS_S1526/27_. ARMS_S1526/27_ showed a significantly increased association with TRPV1. However, even ARMS_0_, where all potential PKA-sites were eliminated, showed a minimal association of 5% with TRPV1, thus indicating an additional PKA-independent factor for interaction. Therefore, forskolin and IBMX were used for maximum PKA stimulation and PKA mediated phosphorylation. While the interaction rate for wild type ARMS (9%) remained the same, stimulation did not significantly increase the MFI association of TRPV1 with ARMS_0_ (6%), ARMS_S1251/52_ (18%) or ARMS_S1526/27_ (63%) and did not significantly decrease the MFI association of ARMS_S882_ (7%), ARMS_T903_ (9%), ARMS_S1439/40_ (6%), and TRPV1, compared to the native untreated condition. This indicated that ARMS is already basal phosphorylated or that PKA mediated phosphorylation does not alter the interaction with TRPV1. Since PKA stimulation did not alter the interaction with TRPV1, the PKA inhibitor H89 was used to induce an interaction change with the potentially basal phosphorylated ARMS. H89 treatment did not decrease the MFI signal rate of ARMS_0_ significantly with TRPV1 to 4% and of ARMS_S1439/40_ to 4% while as well as significantly decreasing the rate of wild type ARMS with TRPV1 to 4%, of ARMS_S882_ to 5%, of ARMS_T903_ to 7%, of ARMS_S1251/52_ to 6% and of ARMS_S1526/27_ to 36_%_. Thus, compared to the condition without PKA inhibitor, the treatment resulted in a significant drop in MFI interaction affecting most mutants and wild type. As the treatment had no significant effect on ARMS_0_, the H89 results indicate a PKA dependent decrease in interaction by inhibiting the basal PKA mediated phosphorylation of ARMS. However, results also indicate an additional factor that enables interaction with TRPV1, as already inferred with untreated ARMS_0_. The results with H89 and untreated ARMS_0_ indicated that an additional factor enables interaction between TRPV1 and ARMS Therefore, the effect of PKD inhibitor BPKDi was determined. Surprisingly, the BPKDi treatment decreased the MFI rate significantly in the wild type and all ARMS mutants: the MFI rate of wild type ARMS with TRPV1 decreased to 3%, the rate of ARMS_0_ to 2%, of ARMS_S882_ to 4%, of ARMS_T903_ to 4%, of ARMS_S1251/52_ to 5%, of ARMS_S1439/40_ to 3%, and of ARMS_S1526/27_ to 25%, resulting in a significant MFI drop in interaction affecting the wild type and all mutants, similar to the effect of H89. The significant drop of ARMS_0_ was especially significant, indicating that the additional factor, enabling interaction between TRPV1 and ARMS, could be PKD mediated phosphorylation.

Subsequently, the phosphorylation status of wild type ARMS and of each ARMS mutant was determined via phospho-PKA substrate antibody (Figure 3). The screening revealed that wild type ARMS and the ARMS mutants are, under native conditions and during H89 or PKD treatment, presumably not phosphorylated by PKA. This refuted the assumption of basal phosphorylated ARMS. Forskolin and IBMX treatment resulted in the phosphorylation of ARMS, TRPV1/ARMS, and TRPV1/ARMS_T903_, while ARMS_0_, ARMS_S882_, ARMS_S1251/52_, ARMS_S1439/40,_ or ARMS_S1526/27_ exposed that they are probably not phosphorylated, even under PKA stimulation. Nevertheless, this shows that ARMS is a target of PKA and that it is phosphorylated at the amino acid T903, even if only stimulated with PKA. Together with the previous results, this leads to the assumption that PKA mediated phosphorylation does not alter the interaction of TRPV1/ARMS, since strong PKA stimulation via forskolin and IBMX did not significantly alter the MFI association of TRPV1/ARMS or its mutants. These results point to a PKA independent interaction.

### 3.3. ARMS/AKAP79 Co-Immunoprecipitation

Since AKAP79 was found to be important in PKA-dependent sensitization of TRPV1 [11], we wanted to explore the role of AKAP79 in the phosphorylation of ARMS. As the phospho-PKA substrate screening revealed, only forskolin and IBMX treatment seemed to result in phosphorylation of ARMS. Therefore, TRPV1 and ARMS were co-expressed in HEK293 cells and stimulated via forskolin (50 µM) and IBMX (100 µM). Cell lysates were verified to contain endogenous AKAP79 by Western Blot. Co-immunoprecipitation experiments were performed with an antibody directed against ARMS. The precipitate was further analyzed by the Luminex FlexMap3D platform (Luminex Corp., Hertogenbosch, The Netherlands) with antibodies against TRPV1, ARMS, AKAP79 (rabbit, D28G3, Cell Signaling, Leiden, The Netherlands), and a phospho-PKA substrate antibody. The screening confirmed the previously determined phosphorylation pattern of ARMS following PKA stimulation with forskolin/IBMX. Furthermore, the screening revealed that AKAP79 is not detectable under the chosen conditions (Figure 4).

To exclude epitope masking, a denaturing Western blot analysis was performed. TRPV1 and ARMS were co-expressed in HEK293 cells and stimulated via forskolin (50 µM) and IBMX (100 µM). Cell lysates were incubated with anti-ARMS (mouse, MA1-90667, Invitrogen, Darmstadt, Germany) coupled Protein G Mag Sepharose and subjected to SDS-PAGE/Western Blot. The membrane was probed with the primary antibodies: anti-ARMS (rabbit, ab34790, abcam, Berlin, Germany), anti-TRPV1 (rabbit, ACC-030, Alomone Labs, Jerusalem, Israel), and anti-AKAP79 (rabbit, D28G3, Cell Signaling, Leiden, The Netherlands). The secondary antibody, anti-Rabbit Alexa Fluor 555 (goat, A-21428, Invitrogen, Darmstadt, Germany), was visualized using Amersham Typhoon Biomolecular Imager (GE Healthcare, Braunschweig, Germany). The Western blot analysis confirmed the previously determined TRPV1 and ARMS interaction (Figure 5). More importantly, AKAP79 seems to be part of the TRPV1/ARMS/PKA signaling complex, regardless of transfection with wild type/mutant ARMS. However, even without transfected TRPV1, AKAP79 and ARMS seemed to interact with each other (Lane 3, ARMS), indicating AKAP79 mediated PKA phosphorylation of ARMS.

### 3.4. Calcium Flux Measurements

To determine potential TRPV1 sensitizing/desensitizing effects of the newly discovered interaction ratios (ARMS/TRPV1 co-immunoprecipitation, Figure 2), TRPV1 and ARMS were co-transfected into HEK293 cells. Calcium flux measurements were performed 48 h post transfection with the fluorescent calcium indicator dye Fluo 4 AM. TRPV1 was activated by capsaicin (0.1 nM–10,000 nM). The calcium flux was determined using a FLUOstar^®^ Omega Microplate Reader (BMG LABTECH, Ortenberg, Germany). The measurements confirmed the known sensitizing effect of ARMS on TRPV1 (Figure 6):

While TRPV1 evinced an EC_50_ of 109 nM for capsaicin, ARMS sensitized TRPV1 significantly to an EC_50_ of 51 nM for capsaicin. ARMS_0_ with an EC_50_ of 54 nM, ARMS_S822_ (EC_50_ of 53 nM), ARMS_T903_ (EC_50_ of 51 nM), and ARMS_S1439/40_ (EC_50_ of 54 nM) seemed to have a similar effect on TRPV1 as wild type ARMS. On the other hand, the sensitizing effect of ARMS_S1251/52_ on TRPV1 was slightly decreased with an EC_50_ of 65 nM. However, ARMS_S1526/27_ showed an even greater sensitizing effect on TRPV1 as wild type ARMS. TRPV1/ARMS_S1526/27_ exhibited an EC_50_ of 26 nM for capsaicin. Together with the results from ARMS/TRPV1 co-immunoprecipitation (Figure 2), this indicates an interaction of rate-dependent sensitization of TRPV1, suggesting that a strong TRPV1/ARMS or mutant ARMS interaction causes a high sensitization of TRPV1. 

For the assessment of the native TRPV1 sensitization, the transfected HEK293 cells were not additionally treated. To assess the influence of PKA mediated ARMS phosphorylation in terms of an increasing/decreasing TRPV1 sensitization, transfected HEK293 were additionally treated with the PKA inhibitor H89 (10 µM), the PKD inhibitor BPKDi (1 µM), or with forskolin (50 µM) combined with IBMX (100 µM). Native condition results from Figure 6 and treated condition results are summarized in Figure 7:

Forskolin and IBMX treatment did not significantly change the calcium flux or the EC_50_ of TRPV1 for capsaicin compared to the native condition, thus, maintaining the known sensitizing effect of ARMS and the newly discovered sensitizing effects of its mutants. On the other hand, H89, as well as BPKDi treatment, significantly decreased the calcium flux of TRPV1, and therefore increased the EC_50_ of TRPV1 when compared to the native condition. Thus, the sensitizing effects of wild type ARMS, ARMS_0_, ARMS_S822_, ARMS_T903_, and ARMS_S1251/52_ and ARMS_S1439/40_ were reversed by H89 or BPKDi treatment. However, the EC_50_ of TRPV1 of single transfected TRPV1 did not significantly alter in the treated conditions compared to the native condition, indicating that the determined sensitization is mediated by ARMS and is independent of TRPV1. While the sensitizing effect of ARMS and its mutants on TRPV1 was reversed by H89 or BPKDi treatment, the ARMS_S1526/27_ mediated sensitization on TRPV1 was not fully reversed. Even under H89 or BPKDi treatment, ARMS_S1526/27_ had a sensitizing effect on TRPV1. Together with the results from ARMS/TRPV1 co-immunoprecipitation (Figure 2 and Figure 3), this indicates an interaction dependent sensitization of TRPV1, where PKA mediated ARMS phosphorylation does not alter the sensitization of TRPV1.

### 3.5. Patch Clamp Experiments

Next, we used electrophysiology to verify our functional calcium imaging results. The co-expression of TRPV1 and wild type ARMS in HEK293 cells induced a significantly increased capsaicin-induced TRPV1 current compared to cells only expressing TRPV1, as previously observed [14]. The sensitizing effect was absent in cells expressing TRPV1 and ARMS_S1251/52_, which is in line with our calcium imaging results. As shown in Figure 8, the co-expression of TRPV1 and ARMS_S1526/27_ resulted in a stronger sensitizing effect on TRPV1 activity compared to wild type ARMS.

### 3.6. shRNA Mediated ARMS Silencing

To verify the involvement of ARMS in TRPV1 sensitization, we silenced ARMS by shRNA. The shC plasmid (ARMS specific) was determined to be the most effective out of four different ARMS targeting shRNA plasmids. The shS plasmid (scrambled control) served as negative control. TRPV1, ARMS, and shRNA plasmids were singly and/or co-transfected into HEK293 cells. Immunoprecipitation experiments (Figure 9) were performed with antibodies directed against ARMS or TRPV1. The precipitate was further analyzed by the Luminex FlexMap3D platform (Luminex Corp., Hertogenbosch, The Netherlands) with antibodies against ARMS for the ARMS precipitate and TRPV1 for the TRPV1 precipitate. The ARMS screening verified the silencing effect of shC on ARMS, which was decreased by a factor of 2.65, while shS showed a nonsignificant and negligible effect on ARMS expression. In addition, the TRPV1 screening showed that neither of the shRNA plasmids had an effect on TRPV1 expression. Thus, indicating that the interaction of TRPV1/ARMS and, respectively, ARMS mediated TRPV1 sensitization could be modulated by shRNA:

To investigate if the shRNA-induced ARMS decrease correlates with a desensitizing effect of TRPV1, TRPV1/ARMS and shRNA plasmids were co-transfected into HEK293 cells. Calcium flux measurements were performed 48 h post transfection with the fluorescent calcium indicator dye Fluo 4 AM. TRPV1 was activated by capsaicin (0.1 nM–10,000 nM). The calcium flux was measured using a FLUOstar^®^ Omega Microplate Reader (BMG LABTECH, Ortenberg, Germany). The measurements confirmed the significantly sensitizing effect of ARMS on TRPV1 (EC_50_ of 109 nM vs. 49 nM), while triple transfection with shC lead to a significantly decreased ARMS-mediated TRPV1 sensitization with an EC_50_ of 91 nM, compared to the TRPV1/ARMS transfection, thus, indicating the reversal of the ARMS-mediated TRPV1 sensitization (TRPV1—TRPV1ARMSshC: not significant). In addition, triple transfection with shS, showed no significant effect (EC_50_ 56 nM) on TRPV1 sensitization, compared to the TRPV1/ARMS transfection. This confirms the assumption that the interaction of TRPV1/ARMS and, respectively, ARMS mediated TRPV1 sensitization can be specifically modulated by shRNA (Figure 10):

## 4. Discussion

In this study we demonstrated that ARMS functions as a PKA substrate in the TRPV1 signaling complex. Previous studies suggested that all three proteins (TRPV1, ARMS, and PKA) interact physically in a signaling complex regulating TRPV1 sensitization in mouse DRG neurons and transfected HEK293 cells [14]. To our knowledge, our study is the first showing PKA phosphorylation of ARMS at peptide and protein level, supporting those previous findings. We show that the PKA-sites in ARMS are phosphorylated in synthetic peptides, in the native protein and partly in ARMS mutants containing a single PKA phosphorylation site. While ARMS_S1251/52_ and ARMS_S1526/27_ are phosphorylated as peptides, they are presumably not phosphorylated in the corresponding mutant proteins, indicating sterical masking of PKA-sites in ARMS. On the other hand, ARMS_T903_ is phosphorylated in both cases.

Beyond providing evidence for ARMS being a PKA substrate, this study quantified MFI-based TRPV1/ARMS interaction rates under the impact of cAMP/PKA pathway stimulation and inhibition. However, the single PKA-site mutants of ARMS revealed that the PKA-mediated phosphorylation does not seem to influence the interaction rate of TRPV1/ARMS. While phosphorylation of ARMS_T903_ does not increase the interaction rate with TRPV1, ARMS_S1526/27_ is probably not phosphorylated and leads to an increased interaction rate. Resulting from these findings and additional experiments using H89 as well as BPKDi, TRPV1/ARMS interaction is independent of PKA phosphorylation of ARMS. This leads to the conclusion that the effects of H89 and BPKDi are either mediated via protein kinase D (PKD) inhibition or via TRPV1 phosphorylation, which implies that phosphorylation of TRPV1 by PKA is required for TRPV1/ARMS interaction on the site of TRPV1. Arévalo et al. showed that ARMS is able to modulate the phosphorylation of the AMPA receptor subunit GluA1 [23]. A more recent study identified ARMS as a modulating interaction partner of voltage-gated sodium channels in the brain [24]. These studies, in addition to ours, underline the ability of ARMS to modulate the activity and phosphorylation status of ion channels.

Furthermore, our data suggest that the TRPV1/ARMS interaction sensitizes TRPV1. Our calcium flux and patch clamp measurements indicate that the higher the interaction rate of TRPV1/ARMS is, the lower is the EC_50_ for capsaicin of TRPV1 and capsaicin-induced TRPV1 currents, independent of the PKA phosphorylation status of ARMS. This demonstrates that the phosphorylation of ARMS_T903_ does not increase the interaction rate with TRPV1 and it does not alter the EC_50_ for capsaicin of TRPV1 compared to wild type ARMS. However, ARMS_S1526/27_ is not phosphorylated and leads to an increased interaction rate, resulting in a decreased EC_50_ for capsaicin of TRPV1 and increased capsaicin-induced TRPV1 current compared to wild type ARMS. The calcium flux measurements in the presence of H89 or BPKDi show that this ARMS mediated TRPV1 sensitization can be reversed, especially as the similar effects of H89 and BPKDi on ARMS_0_ support the conclusion that this ARMS mediated TRPV1 sensitization is dependent on PKD phosphorylation. Additionally, the assumption, that ARMS_S1526/27_ is not phosphorylated by PKA leads to the conclusion that the H89 effect is either mediated via TRPV1 phosphorylation or like in the case of ARMS_0_, via inhibition of protein kinase D (PKD) and completely independent of PKA phosphorylation of ARMS. Thus, interaction of ARMS and TRPV1 leads to the sensitization of TRPV1, but is independent of PKA phosphorylation of ARMS, and seems to be dependent on PKA phosphorylation of TRPV1 or PKD phosphorylation of ARMS. Substantial interaction of TRPV1 and PKA has already been demonstrated by others. While PKA phosphorylation sites of TRPV1 have been established, further studies are required to identify PKA relevant TRPV1/ARMS interaction sites [7,25]. For PKA it was shown that it is involved in the insertion of functional TRPV1 tetramers into the plasma membrane by forming a complex of AKAP79, PKA, and TRPV1 [26]. It was shown that AKAP79 can arrange TRPV1, adenylyl cyclase and PKA to form a sophisticated complex vital for TRPV1 sensitization [27,28]. Our study is the first showing that AKAP79 is also part of the TRPV1/ARMS/PKA signaling complex. However, even without TRPV1 being present, AKAP79 and ARMS seem to interact and form a complex. By showing physical and functional interactions between TRPV1, ARMS, PKA, and AKAP79, our data suggest that all four proteins are part of a sophisticated signaling complex that sensitizes TRPV1 towards capsaicin.

Since the TRPV1-sensitizing effect of the ARMS/PKA/AKAP79 complex seems to be independent of PKA phosphorylation, but dependent on interaction with ARMS, a drastically reduced interaction should stop the sensitization of TRPV1. To overcome the problem of ARMS-mediated TRPV1 sensitization by interaction, we silenced ARMS by shRNA. ARMS silencing resulted in a restored TRPV1 desensitization without affecting the TRPV1 expression, and therefore could be used as new topical therapeutic analgesic alternative to stop ARMS mediated TRPV1 sensitization.

## 5. Conclusions

Our study is the first showing PKA phosphorylation of ARMS at peptide and protein level. However, single PKA-site mutants of ARMS revealed that the PKA-mediated phosphorylation of ARMS did not alter the interaction rate of TRPV1/ARMS and sensitization of TRPV1, respectively. Yet, we demonstrate that ARMS is a PKA substrate tethered via AKAP79 into the TRPV1 signaling complex and that all four proteins interact physically, regulating TRPV1 sensitization in transfected HEK293 cells.

## Figures and Tables

**Figure 1 medsci-10-00063-f001:**
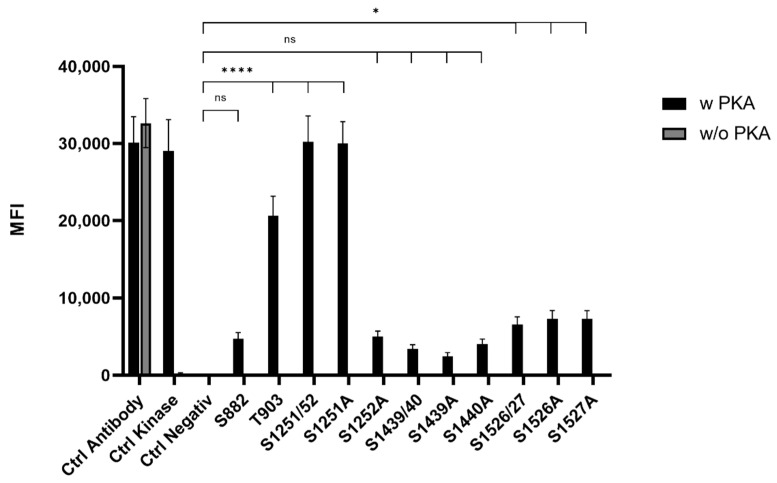
Mean fluorescence intensity (MFI) histogram of PKA phosphorylated sites in ARMS. Data are presented as means ± SEM. Statistical significance is denoted as ns = not significant, *p* < 0.05 (*), *p* < 0.0001 (****). *n* = 3, biological replicates. BLACK: PKA phosphorylated site, GREY: Phospho-PKA substrate antibody positive control.

**Figure 2 medsci-10-00063-f002:**
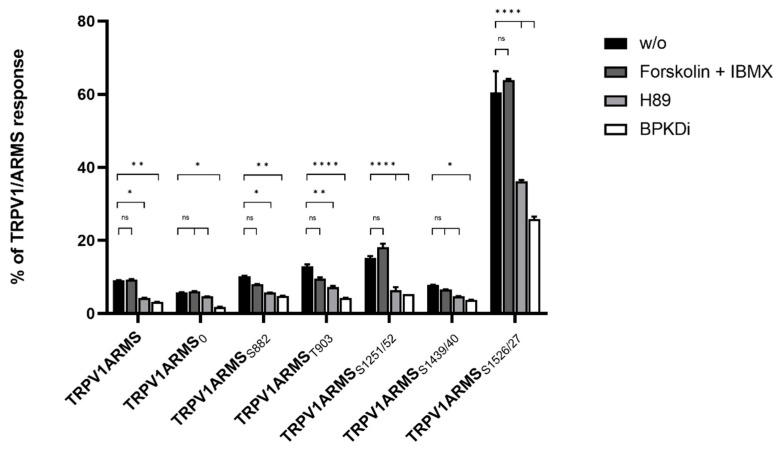
Co-immunoprecipitation of TRPV1/ARMS in transfected HEK293. Histogram of TRPV1/ARMS response in %. Data are presented as means ± SEM. Statistical significance is denoted as ns = not significant, *p* < 0.05 (*), *p* < 0.01 (**), *p* < 0.0001 (****). *n* = 3, biological replicates. Interaction ratio was calculated and plotted: TRPV1 divided by ARMS (MFI). BLACK: without stimulation or inhibition, DARK GREY: PKA stimulated with forskolin and IBMX, GREY: PKA inhibited with H89, WHITE: PKD inhibited with BPKDi.

**Figure 3 medsci-10-00063-f003:**
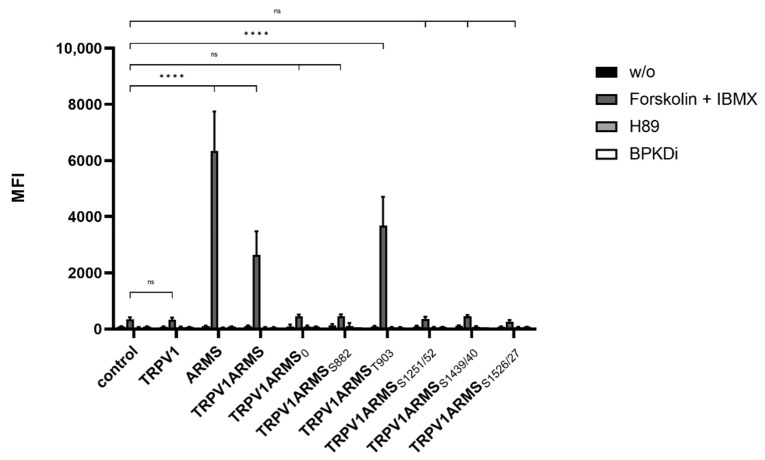
Histogram of the phosphorylation status in single and double TRPV1/ARMS transfected HEK293 via phosphor-PKA substrate antibody in MFI. Data are presented as means ± SEM. Statistical significance is denoted as ns = not significant and *p* < 0.0001 (****). *n* = 3, biological replicates. BLACK: without stimulation or inhibition, DARK GREY: PKA stimulated with forskolin and IBMX, GREY: PKA inhibited with H89, WHITE: PKD inhibited with BPKDi.

**Figure 4 medsci-10-00063-f004:**
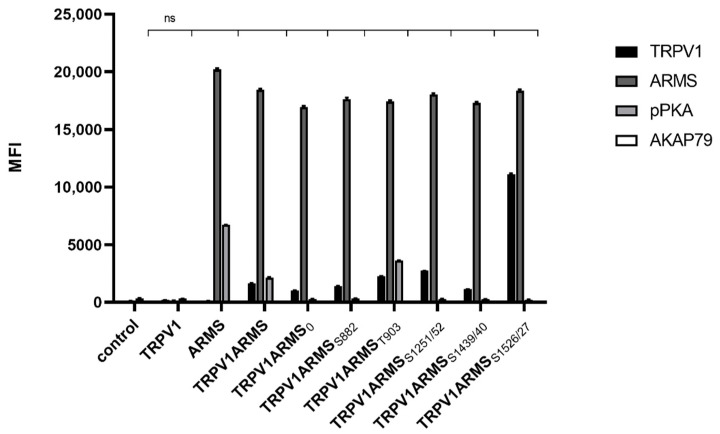
Co-immunoprecipitation of ARMS/AKAP79 in TRPV1/ARMS transfected and PKA stimulated HEK293 via anti-TRPV1, anti-ARMS, anti-AKAP79, and phosphor-PKA substrate antibody in MFI. Data are presented as means ± SEM. Statistical significance is denoted as ns = not significant. *n* = 3, biological replicates. BLACK: TRPV1, DARK GREY: ARMS, GREY: pPKA, WHITE: AKAP79.

**Figure 5 medsci-10-00063-f005:**
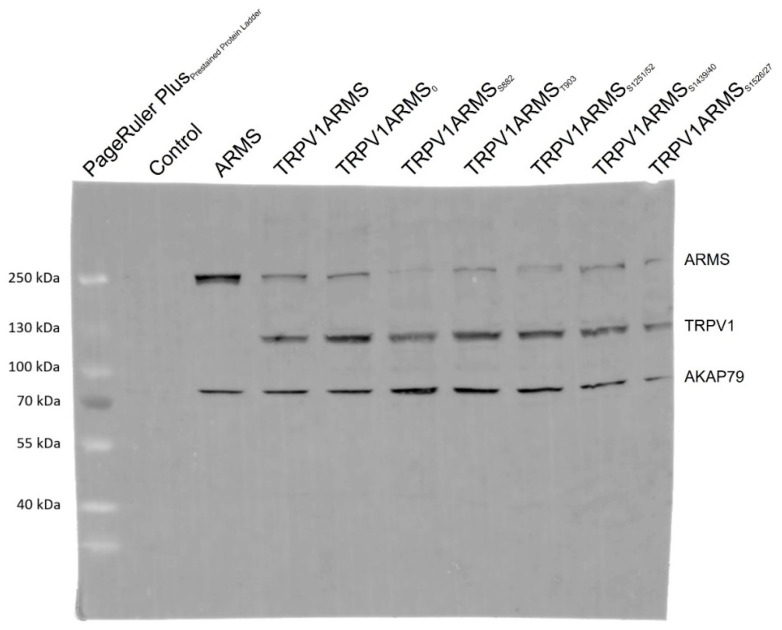
Western blot analysis of TRPV1/ARMS/AKAP79 in TRPV1/ARMS transfected and PKA stimulated HEK293 via anti-TRPV1, anti-ARMS, and anti-AKAP79. BANDS at ~79 kDa: AKAP79, BANDS at ~120 kDa: TRPV1, BANDS at ~220 kDa: ARMS.

**Figure 6 medsci-10-00063-f006:**
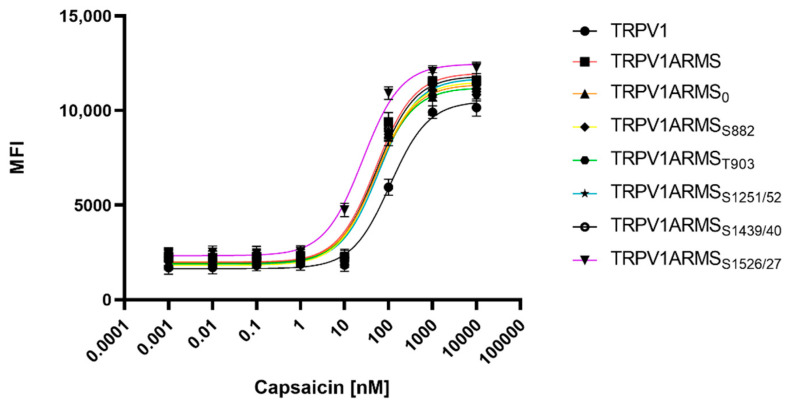
Calcium flux dose response curves of TRPV1/ARMS in transfected HEK293. MFI Data was plotted against capsaicin concentration and is presented as means ± SEM. Dose–response curves were fitted with the Hill equation.

**Figure 7 medsci-10-00063-f007:**
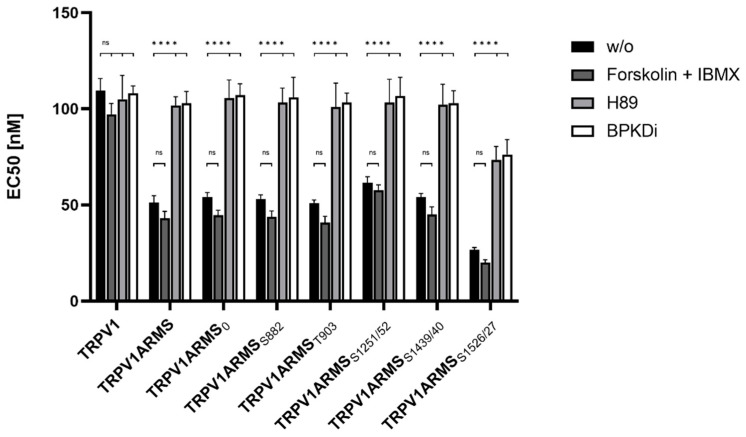
EC_50_ histogram of TRPV1/ARMS in transfected HEK293. Data are presented as means ± SEM. Statistical significance is denoted as ns = not significant and *p* < 0.0001 (****). *n* = 3, biological replicates. BLACK: without stimulation or inhibition, DARK GREY: PKA stimulated with forskolin and IBMX, GREY: PKA inhibited with H89, WHITE: PKD inhibited with BPKDi.

**Figure 8 medsci-10-00063-f008:**
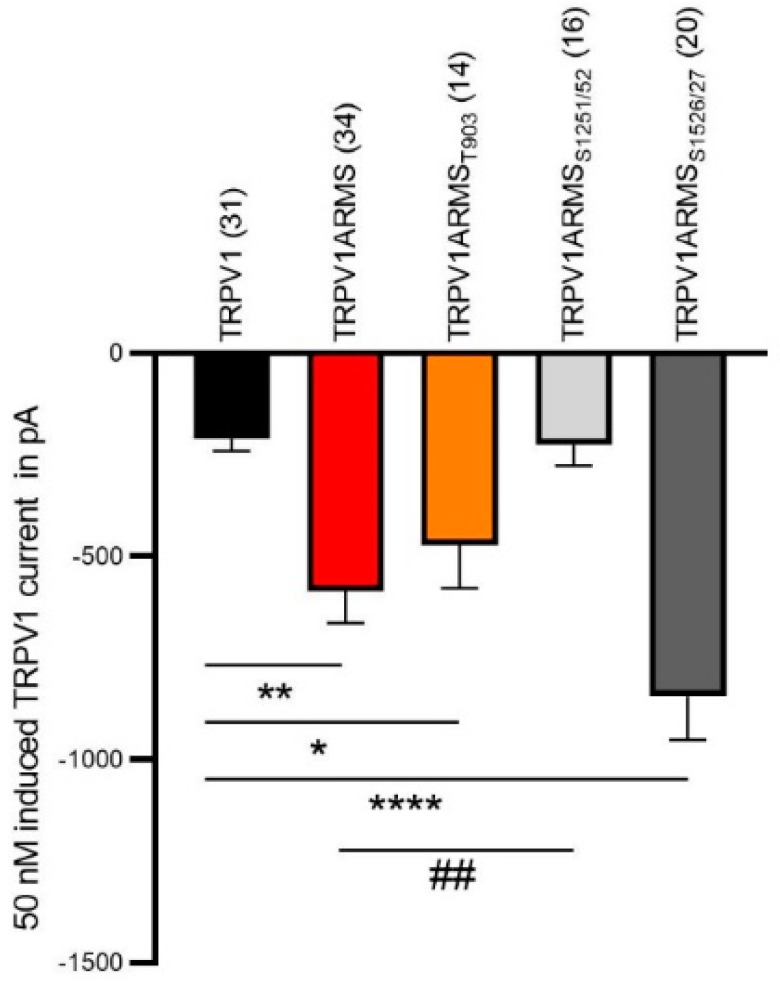
Capsaicin-induced (50 nM) TRPV1 current (in pA) of HEK cells expressing TRPV1. TRPV1 (black column, *n* = 31 cells), TRPV1 and wild type ARMS (red, *n* = 34 cells), TRPV1 and ARMS_T903_ (orange, *n* = 14 cells), TRPV1 and ARMS_S1251/52_ (light grey, *n* = 16 cells) and TRPV1 ARMS_S1526/27_ (dark grey, *n* = 20 cells). Data were analyzed using the Kruskal-Wallis test with Dunn’ multiple comparing test. *, **, **** indicates TRPV1 as comparison control group, ## indicates TRPV1ARMS as the comparison control group in the post-test.

**Figure 9 medsci-10-00063-f009:**
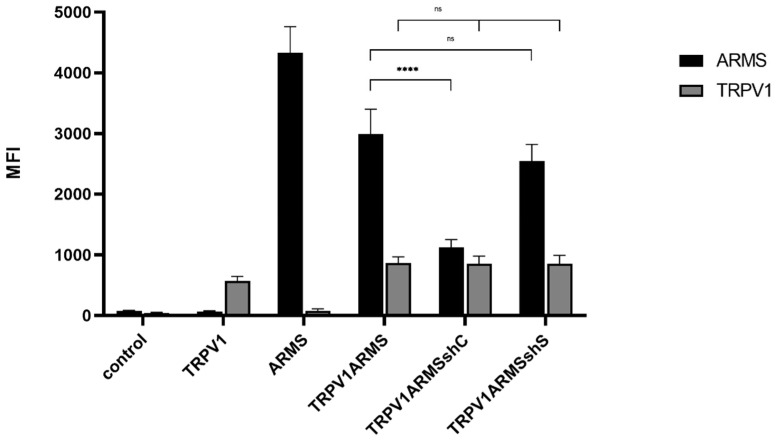
Immunoprecipitation of ARMS and TRPV1 in transfected HEK293. Histogram of ARMS and TRPV1 response in MFI. Data are presented as means ± SEM. Statistical significance is denoted as ns = not significant and *p* < 0.0001 (****). *n* = 3, biological replicates. BLACK: ARMS, GREY: TRPV1.

**Figure 10 medsci-10-00063-f010:**
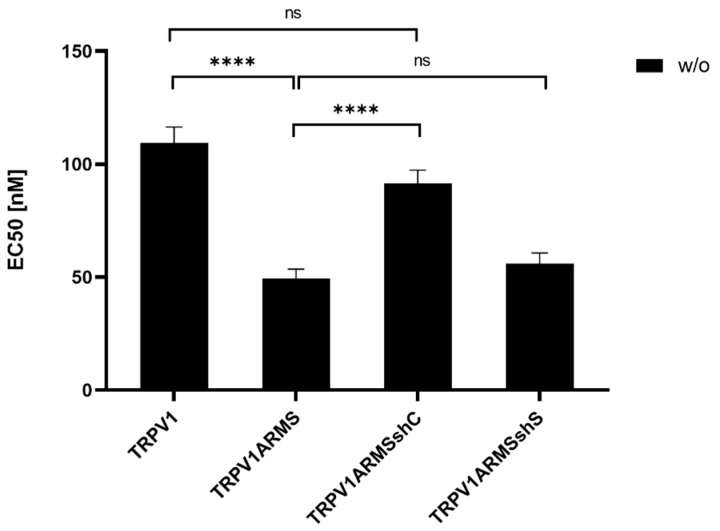
EC50 histogram of TRPV1/ARMS in shRNA transfected HEK293. Data are presented as means ± SEM. Statistical significance is denoted as ns = not significant and *p* < 0.0001 (****). *n* = 3, biological replicates. BLACK: without stimulation.

**Table 1 medsci-10-00063-t001:** Plasmids.

	Sequence
TRPV1 [21] (UniProtKB-O35433)	pcDNA 3.1 rTRPV1-YFP
ARMS [22] (UniProtKB-Q9EQG6-2)	pcDNA 3.1 rARMS, S131T, Q136^-^
ARMS_0_	pcDNA 3.1. rARMS, S882A, T930A, S1251/52A, S1439/40A, S1526/27A
ARMS_S882_	pcDNA 3.1. rARMS, T930A, S1251/52A, S1439/40A, S1526/27A
ARMS_T903_	pcDNA 3.1. rARMS, S882A, S1251/52A, S1439/40A, S1526/27A
ARMS_S1251/52_	pcDNA 3.1. rARMS, S882A, T903A, S1439/40A, S1526/27A
ARMS_S1439/40_	pcDNA 3.1. rARMS, S882A, T930A, S1251/52A, S1526/27A
ARMS_S1526/27_	pcDNA 3.1. rARMS, S882A, T903A, S1251/52A, S1439/40A
shC (ARMS)	pRFP-C-RS, 5′ CTGTTACTGAGTTCAATGACCGTGGATGT 3′
shS (scamble)	pRFP-C-RS, 5′ GCACTACCAGAGCTAACTCAGATAGTACT 3′

**Table 2 medsci-10-00063-t002:** Peptides.

	Sequence
S882	HGTQEDTDRRVSQNSLGGKPIPNPLLGLDST
T903	HGSKTALNRRDTYRRRQGKPIPNPLLGLDST
S1251/52	HVPHGESARRSSHTELPGKPIPNPLLGLDST
S1251A	HVPHGESARRASHTELPGKPIPNPLLGLDST
S1252A	HVPHGESARRSAHTELPGKPIPNPLLGLDST
S1439/40	HGSKLLPGKKSSERPSLGKPIPNPLLGLDST
S1439A	HGSKLLPGKKASERPSLGKPIPNPLLGLDST
S1440A	HGSKLLPGKKSAERPSLGKPIPNPLLGLDST
S1526/27	HYLSDALLDKKDSSDSGVRGKPIPNPLLGLDST
S1526A	HYLSDALLDKKDASDSGVRGKPIPNPLLGLDST
S1527A	HYLSDALLDKKDSADSGVRGKPIPNPLLGLDST

## Data Availability

The data presented in this study are available on request from the corresponding author.

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
