# Peer review of "Transient Receptor Potential Vanilloid 1 Signaling Is Independent on Protein Kinase A Phosphorylation of Ankyrin-Rich Membrane Spanning Protein"

_medsci, 2022, doi:10.3390/medsci10040063_

Round 1
Reviewer 1 Report
Pellegrino et al performed in-depth analysis on TRPV1 signaling where ARMS is a PKA substrate via AKA79. I have moonier comments as follows:
1. Fig 5: Full western blot is missing. Protein in last lane not presented well.
2. All the co-immunoprecipitation results of TRPV1/ARMS are shown in graphs. Please show a representative image of western blotting, showing the co-IP with antibody control lane.
Reviewer 2 Report
The following points need correction and/or further explanation or comments by the authors.
1. Page 1 line 19: “Additionally, experiments were done to discriminate between PKA and PKD …”
Please provide explanation (full name) for abbreviation of PKD in the abstract as you have done with all other proteins.
2. Page , line 113: “AKA79 was also involved in the TRPV1/ARMS/PKA signaling complex
Please provide explanation (full name) for abbreviation of AKA79 in the abstract as you have done with all other proteins.
3. Page 4, line: “ARMS peptides containing possible PKA-sites” ARMS peptides are commercial available ARMS peptides how authors can be reassured that those peptides are similar to ARMS expressed by HEK293 transfected cells (pcDNA3.1-ARMS page 2)? The ARMS expressed in HEK293 may/possible has posttranslational modification lacking by those ARMS peptides. Those modifications may effect binding to PKA and/or thus phosphorylation. What is the benefit of this experiment since authors have transfected HEK293 with ARMS mutants (figure 2) and thus study directly the mutant protein rather a sort (peptide) part of the protein?
4. Page 4, Table 1: “S1251/52A, S1439/40A, S1526/27A”. The residue 1251, 1439 and 1526 (see sequence https://www.uniprot.org/uniprotkb/Q9EQG6/entry) are not serine (S) residues as authors descripted but M1251, R1439, T1526 for isoform Q9EQG6-1 and R1251, K1439, D1526 for isoform Q9EQG6-2 respectively.
Similar errors/typo are present for all mutants of ARMS on table 1.
There is probable a typo since for isoform Q9EQG6-2 indeed the residues 1252 and 1253 are serine but not residues 1251 and 1252 as authors mentioned in the manuscript. The same for S1439/40 should be corrected to S1440/41
5. Page 4, Table 2: sequence S882 (HGTQEDTDRRVSQNSLGGKPIPNPLLGLDST) do not match 100% with sequence https://www.uniprot.org/uniprotkb/Q9EQG6/entry. From 32 residues peptides 17 residues (central core) match the rest do not. Please explain why authors do not use a sequence that matches all peptides of ARMS .
The same apply for the rest of the sequences presented in Table 2.
According to sequence https://www.uniprot.org/uniprotkb/Q9EQG6/entry for peptide S882 (RRVSQ) the residues S corresponds to residue 883 and not 882.
6. Page 5 lines Figure 1: Authors used anti-Phospho-PKA substrate antibody. (rabbit, 100G7E, Cell Signaling). This antibody is specific for substrate RRXS*/T*. Analysis of Signal-to-noise data for similar substrates are shown in Cell Signalling website (https://www.cellsignal.com/products/primary-antibodies/phospho-pka-substrate-rrxs-t-100g7e-rabbit-mab/9624?site-search-type=Products&N=4294956287&Ntt=phospho-pka+substrate&fromPage=plp ). In those data they use similar peptides to current publication and PKA as kinase enzyme. According to above information the best epitope site (RRXS*/T*) should be S1251/52, S1251A, S882, T903 following by the rest of peptides on Table 2 and the lower signal should come from peptides S1252A, S1440A and S1526/27. Indeed, authors observed the higher signal/noise ratio for S1251/52 and S1251A but not for S882 could authors comment why?
S882 sequence matches well with epitope RRXS*/T* and with sequence of Synapsin (Ser9) in the above website that gives a S/N ratio of 50% when in current manuscript the is approximately 11% of higher S/N ratio.
7. Page 6, lines 210-211: “However, even ARMS0, where all potential PKA-sites were eliminated, showed a minimal association of 5 % with TRPV1. Thus, indicating an additional PKA-independent factor for interaction”
This is a strong statement by authors. The ARMS0 showed a minimal association 5% but ARMSS1526/27 a significant association 63 %. The ARMSS1526/27 indicating an additional PKA-dependent factor for interaction which is further confirmed by the addition of inhibitor BPKDi and the significantly decreased (by 2/3) association with TRPV1.
8. Page 6, lines 218-219: “Thus, indicating that ARMS is already basal phosphorylated or that PKA mediated phosphorylation does not alter the interaction with TRPV1”
It could by equally possible that that wild type unphosphorylated ARMS is interacts with TRPV1. In order to confirm that ARMS is basal phosphorylated authors can digest ARMS (Dephoure N, et al., Mapping and analysis of phosphorylation sites: a quick guide for cell biologists. Mol Biol Cell. 2013, 5, 535-42) and run a mass analysis in order to identify phosphorylate sites (Mass spectrometry). This is a simple routinely experiment that can run in many service facilities. Another way to confirm the above statement could simple use a phosphatase (see FastAP Thermosensitive Alkaline Phosphatase from Thermo Fisher Scientific) that removes phosphate groups from proteins and run the Co-immunoprecipitation of TRPV1/ARMS afterwards.
9. Page 6, lines 229-230: “However, results also indicate an additional factor that enables interaction with TRPV1, as already inferred with untreated ARMS0”
I will not argue that there is an additional factor that enables interaction with TRPV1, as already inferred with untreated ARMS0. Although this additional factor seems to be of low significant. The experiment of TRPV1/ARMS0 (figure 2) have the lowest Statistical significance (p<0.05) and the lowest MFI on the other hand the TRPV1/ARMSS1526/27 has the highest Statistical significance (p<0.001) and the highest MFI and indicates a strong interaction with TRPV1 due to phosphorylated and thus strong interaction with treated ARMS.
10. Page 6 line 232: “Surprisingly, the BPKDi treatment decreased the MFI rate significantly in the wild type and all ARMS mutants” It may be not a surprise if authors consider the possibility that there are additional PKA phosphorylated sites in ARMS that they have not included in the Peptides investigated in Table 2. It is also possible those PKA phosphorylated sites may be not specific for the anti-Phospho-PKA substrate antibody (rabbit, 100G7E, Cell Signaling) used in the current manuscript.
11. Page 7 line 240-241: “Subsequently, the phosphorylation status of wild type ARMS and of each ARMS mutant was determined via phospho-PKA substrate antibody (Figure 3)”
Please provide a sort description (in material and method section) how this experiment was performed
12. Page 7 line 241-243: “The screening revealed that wild type ARMS and the ARMS mutants are, under native conditions and during H89 or PKD treatment, presumably not phosphorylated by PKA”
This is a solid/major evident that wild type ARMS under native conditions is not phosphorylated by PKA. I will suggest the manuscript title to change from “Transient receptor potential vanilloid 1 signaling is independent on Protein kinase A phosphorylation of Ankyrin-rich membrane spanning protein” to “Transient receptor potential vanilloid 1 signaling is independent under native conditions on Protein kinase A phosphorylation of Ankyrin-rich membrane spanning protein”
13. Page 7 line 246-247: “ARMSS1526/27 exposed that they are probably not phosphorylated, even under PKA stimulation” the antibody used (see reviewer comment 7) is not so specific for ARMSS1526/27 this can be seen also from experimental results in figure 1 where signal/noise ratio for S1526/27 has low MFI and the lowest Statistical significance (p<0.05), please comment
14. Page 7 figure 3: The pattern of MFI in figure 3 and figure 1 are not similar although the peptides used in figure 1 and the proteins in figure 3 have the same epitopes. The most profound differences are those of S1251/52 and S1252 where in figure 1 demonstrates the highest MFI of all mutant but in figure 3 the opposite (the second lowest MFI). Equally surprisingly the S1526A has the lowest MFI in figure 3 but in figure demonstrates the forth MFI from the top.
15. Page 8 line 290-292: “However, even without transfected TRPV1, AKAP79 and ARMS seemed to interact with each other, indicating AKAP79 mediated PKA phosphorylation of ARMS”
Is that evident from figure 5? Please indicate lanes in SDS-page.
If that derives from previous data please add reference
16. Page 10 line 316-317: “this indicates an interaction rate dependent sensitization of TRPV1. Meaning, a strong TRPV1/ARMS or mutant ARMS interaction causes a high sensitization of TRPV1”.
In otherworld the bars (w/o and Forskolin and IBMX) in figure 7 should be reciprocal of those in figure 2. This is true for mutant ARMSS1526/27 but the same cannot be said for ARMSS1251/52 and ARMSS1539/40. The last two mutants should have lower EC50 than all other mutants except ARMSS1526/27
17. Page 13 line 406-407: “While ARMSS1251/52 and ARMSS1526/27 are phosphorylated as peptides, they are presumably not phosphorylated in the corresponding mutant proteins, indicating structural masking of PKA-sites in ARMS.”
If the above is correct how authors explain the strong influence of phosphorylation due to addition of PKD inhibitor BPKDi in figure 2.
Round 2
Reviewer 2 Report
The revised manuscript and the cover letter have addressed all comments.